# Breath-Counting Task Enhances the Sensitivity of Fear Acquisition

**DOI:** 10.3390/bs15030263

**Published:** 2025-02-24

**Authors:** Xu Li, Yong Yang, Ranran Wang, Lehong Zhou, Xifu Zheng

**Affiliations:** 1School of Humanities and Law, Guangdong University of Petrochemical Technology, Maoming 525000, China; psyxl525@163.com (R.W.); zhoulehong@gdupt.edu.cn (L.Z.); 2School of Educational Science, Xinyang Normal University, Xinyang 466000, China; yangyongseeklight@m.scnu.edu.cn; 3Key Laboratory of Brain, Cognition and Education Sciences, Ministry of Education, South China Normal University, Guangzhou 510631, China; zhengxifu@m.scnu.edu.cn; 4School of Psychology, Center for Studies of Psychological Application, Guangdong Key Laboratory of Mental Health and Cognitive Science, South China Normal University, Guangzhou 510631, China

**Keywords:** conditioned fear, fear acquisition, mindfulness, breath-counting task

## Abstract

Fear acquisition is an essential survival mechanism for humans; however, the role and mechanisms of mindfulness training in this process remain unclear. This study employed a discriminative fear conditioning paradigm to investigate the effects and mechanisms of short-term mindfulness training, exemplified by the breath-counting task, on fear acquisition. The experiment consisted of three consecutive phases: intervention, habituation, and acquisition. During the intervention phase, each participant was assigned to one of two conditions: the breath-counting task group (experimental group) or the free reading group (control group). The results indicated that the mindfulness group exhibited significantly lower expectancy ratings for shocks to the CS− compared to the control group, while no significant difference was found in the shock ratings for CS+. Regarding skin conductance responses, although the mindfulness group showed a significantly reduced fear response to CS relative to the free reading group, there was no significant difference in overall fear acquisition effects between the two groups. The above findings indicate that breath-counting tasks enhance sensitivity to the acquisition of conditioned fear by reducing exaggerated fear responses to safety signals. The conclusions of this study further elucidate the conflicting results regarding the effects of mindfulness training on fear acquisition and provide novel perspectives for the prevention of anxiety spectrum disorders.

## 1. Introduction

Fear learning enables individuals to distinguish between threat and safety signals, allowing them to quickly anticipate and respond to dangerous environments—an essential process for survival, development, and mental health ([30]). Impaired signal discrimination during fear learning, particularly when safety or neutral cues are incorrectly perceived as threats, can deplete substantial attentional resources and may lead to the onset of anxiety spectrum disorders ([1]; [36]). Thus, improving signal discrimination in fear learning is crucial for preventing psychological disorders and supporting mental health ([33]; [38]).

Mindfulness involves accepting present experiences with an open, non-judgmental attitude ([27]). Since Kabat-Zinn introduced mindfulness from its Buddhist roots into psychotherapy, mindfulness has become widely researched and applied due to its demonstrated efficacy and compatibility with existing clinical methods ([10]). Mindfulness training is typically categorized into short-term and long-term formats. Short-term training encompasses various forms, such as mindfulness breathing exercises, body scanning, and meditation, with duration ranging from a single session of 15 to 30 min to approximately two weeks ([45]). Long-term training typically follows structured programs like Mindfulness-Based Cognitive Therapy (MBCT) or Mindfulness-Based Stress Reduction (MBSR), lasting several weeks to months ([2]). Although debates continue in academia concerning the precise definition of mindfulness, most researchers concur that mindfulness training can improve present-moment interoception ([17]) and improve individuals’ attentional control ([50]) and emotional regulation skills ([18]; [22]). This helps individuals avoid avoidance behaviors and internal reactions, fostering greater acceptance of bodily and emotional responses ([34]). Cognitive neuroscience research has demonstrated that mindfulness training promotes an increase in gray matter within the hippocampus ([43]). This enhancement strengthens the connections between the hippocampus and the primary sensory cortex, leading to optimized selective attention to sensory information and improving individuals’ accuracy in perceiving sensory stimuli ([46]). Furthermore, studies indicate that mindfulness meditation can regulate amygdala function through the functional connectivity of the medial prefrontal cortex, thereby facilitating emotional regulation ([28]).

In recent years, there has been a growing interest among researchers in the impact of mindfulness on the process of fear acquisition and fear extinction. Studies showed that mindfulness training results in reduced reinstatement and enhanced extinction effects compared to control groups. These results indicated that mindfulness could potentially emerge as a preferred treatment for mitigating anxiety and its associated symptoms ([39]). Mindfulness practice seems to protect individuals from reacquiring conditioned fear after extinction ([26]) and enhances neural connectivity when retrieving extinguished fear memories ([46]). Research on the influence of mindfulness on fear acquisition remains limited. [23] ([23]) employed a pre- and post-test experimental design to investigate the effects of mindfulness training on the acquisition and extinction of fear, as well as to assess whether changes in white matter fiber bundles could facilitate this transformation. Their study revealed that, after eight weeks of mindfulness training, participants displayed a significant interaction relating to the differences between the conditioned stimuli CS+ and CS−, indicating ongoing sensitivity to the conditioned stimuli during the fear acquisition phase subsequent to mindfulness training. Similarly, [4] ([4]) examined the impact of mindfulness training on conditioned fear acquisition and extinction through a two-day conditioning experiment. They found that, after four weeks of mindfulness training via a smartphone application, no notable differences were observed between the mindfulness group and the control group in terms of fear acquisition and extinction on the first day of the experiment. However, the mindfulness group exhibited a reduced spontaneous recovery of threat-related arousal responses on the second day. In contrast, another study employing a delayed eyeblink conditioning protocol found that after three weeks of repeated mindfulness training, mindfulness training interfered with classical conditioned responses, delaying the onset of the first conditioned response and decreasing the frequency of conditioned eyeblinks ([19]). The discrepancies observed in these findings may be partially attributed to the interference of extraneous factors during prolonged mindfulness training and could also be linked to variations in experimental designs or response metrics. Therefore, subsequent research must further elucidate and clarify the effects of mindfulness training on conditioned fear acquisition, considering both mindfulness training methodologies and experimental designs.

Long-term mindfulness training presents significant challenges to both its effectiveness and the control of confounding factors in experimental settings, primarily due to the protracted duration required. Furthermore, individuals today navigate a fast-paced modern life, which limits the feasibility of long-term mindfulness programs. These programs are often associated with substantial time commitments and elevated dropout rates ([14]). Consequently, it is essential to investigate the role and mechanisms of short-term mindfulness training in the process of fear acquisition. [31] ([31]) introduced the breath-counting task (BCT) as an effective tool for assessing mindfulness behavior. This task involves meditation techniques centered on breath awareness, requiring participants to concentrate on their breathing and accurately count their breaths, with counting accuracy serving as a proxy for mindfulness levels. As a single-session, short-duration mindfulness intervention, BCT not only evaluates the effectiveness of participants’ mindfulness practice ([24]) but also specifically reflects the working mechanisms of mindful breathing in attention and emotional regulation ([25]). In other words, BCT aids participants in maintaining present-moment awareness through mindful breath counting, fostering a direct experience of bodily sensations. This, in turn, promotes an open and accepting attitude ([7]; [55]), potentially attenuating negative emotions ([8]).

The discriminative fear conditioning paradigm is a crucial behavioral model for investigating fear memory processes, including learning, storage, retrieval, and extinction, as well as the central mechanisms underlying these processes ([3]). This paradigm simulates how humans differentiate between threatening and non-threatening stimuli in everyday life and develop defensive responses necessary for survival ([15]). In the fear acquisition phase, one stimulus (CS+) is repeatedly associated with an aversive stimulus, such as a shock, while another stimulus (CS−) is consistently not followed by the shock. Consequently, individuals exhibit a fear response to the CS+ in a manner comparable to that triggered by the shock, whereas the CS− serves as a safety signal, preventing a fear response. In the discriminative conditioned fear paradigm, two commonly used response indicators are unconditioned stimulus (US) expectancy and skin conductance response (SCR). US expectancy reflects the participants’ level of subjective anticipation regarding fear stimuli, while SCR indicates the physiological arousal level of participants in response to fear stimuli ([51]). These two types of fear response indicators are extensively employed in clinical psychology to explore the psychopathological and neurophysiological foundations of anxiety-related disorders.

In conclusion, BCT serves not only as an objective tool for measuring mindful behavior but also as an effective method for short-term mindfulness training. However, no research to date has explored the role and mechanism of BCT in the acquisition of conditioned fear. This study investigated the role of BCT in the acquisition of conditioned fear, employing US expectancy and SCR as indicators of fear responses. We hypothesized that this training may enhance individuals’ capacity to recognize safe and dangerous signals during the process. The findings may further clarify the conflicting results concerning the impact of mindfulness training on fear acquisition while offering new insights into the prevention of anxiety spectrum disorders.

## 2. Materials and Methods

### 2.1. Participants

This study employed a 2 × 2 mixed experimental design, and the sample size was estimated using GPower (*α* = 0.05, 1 − *β* = 0.8, *f* = 0.25), resulting in a total of 34 participants. In the current study, 70 college students voluntarily participated in the experiment, recruited through an online platform, and were randomly assigned to either the experimental group or the control group. Two participants were excluded due to equipment malfunctions, and an additional five participants were removed because of insufficient attentiveness in their responses ([55]). Consequently, the statistical analysis included data from 63 participants (33 males and 30 females) aged between 18 and 26 years (*M* = 19.78, *SD* = 1.59).

All participants exhibited normal corrected vision, had no history of mental disorders, and reported not having taken any psychotropic medications in the recent past. Prior to the commencement of the experiment, participants were informed of their rights during the experimental process and were required to sign a written informed consent form. Additionally, participants who successfully completed the experiment would receive compensation. This study received approval from the Academic Ethics Committee of the Guangdong University of Petrochemical Technology and ensured compliance with the ethical requirements stipulated in the Helsinki Declaration.

### 2.2. Materials

Consistent with the methodology outlined by [35] ([35]), this study utilized two circles with different diameters (5.00 cm and 11.75 cm) as conditioned stimuli (CS). The circles were designated as CS+ and CS−, and their respective roles were counterbalanced among various participant groups. In alignment with the existing literature (e.g., [49]), shock stimulation was employed as the unconditioned stimulus (US), administered via a constant-voltage stimulator (DS2A, Digitimer Ltd., Hertfordshire, UK). CS+ was associated with a 75% probability of being paired with a shock, whereas CS− was never paired with a shock. Each electrical stimulus lasted for 200 ms, and the intensity was meticulously calibrated for each participant to ensure it was perceived as “extremely uncomfortable but bearable” before the experiment commenced.

The State–Trait Anxiety Inventory Form Y ([32]; [48]) comprises 20 items for each subscale and utilizes a 4-point Likert scale, in which higher scores indicate greater levels of state or trait anxiety among participants. In this study, the internal consistency for the state and trait anxiety subscales was 0.94 and 0.93, respectively. The Mindful Attention Awareness Scale ([5]; [6]) includes 15 items and employs a 6-point Likert scale, where higher total scores reflect increased levels of trait mindfulness. The internal consistency of this scale was 0.88 in the current study.

### 2.3. Measures

#### 2.3.1. US-Expectancy Ratings

During the acquisition phases, a prompt reading, “What is the likelihood of a shock occurring afterward?” was displayed beneath the stimuli. Participants were instructed to respond by pressing a key from 1 to 5, with higher numbers indicating a stronger subjective belief in the expectancy of a shock.

#### 2.3.2. Skin Conductance Responses

SCRs were recorded at a frequency of 1000 Hz using a physiological recording device (BIOPAC MP160) and subjected to baseline correction for data reduction with AcqKnowledge 5.0 software. Specifically, the raw SCRs induced by the CS were calculated by subtracting the average SCRs from the 2 s prior to the presentation of the CS from the peak SCRs observed within 5 s after the presentation of the CS ([47]). All raw SCR data from the participants must undergo range correction, with values below 0.02 μs adjusted to 0. Furthermore, all SCR data underwent square root transformation to mitigate distribution skewness ([44]).

### 2.4. Procedure

The experimental procedure was implemented using E-Prime 3.0. According to [52] ([52]), the experiment consists of three sequential phases: intervention, habituation, and acquisition. During the habituation and acquisition phases, participants kept the ring and index fingers of their left hand connected to a physiological sensor while connecting the constant-voltage stimulator with their right hand. Throughout the experiment, participants used a numeric keypad to respond, also with their right hand. The specific details of each phase are as follows:

Intervention Phase: This phase consisted of two groups. The experimental group performed a 15 min BCT. Participants were instructed to assume a relaxed posture in front of a computer screen and were asked to monitor their respiration by counting their breaths while maintaining a natural breathing pattern. One inhalation and one exhalation were counted as one breath, with participants responding via key presses. Each cycle comprised 9 breaths: the first 8 breaths were registered by pressing the “down arrow”, and the 9th breath by pressing the “right arrow”. In the event of a counting error, participants pressed the “space bar” to initiate the next cycle immediately. Throughout the task, participants were required to maintain focus and keep their eyes open. Counting accuracy is defined as the percentage of correct breath-counting cycles in relation to the total number of cycles. Following [55] ([55]), participants with a counting accuracy below 7% were considered to have not seriously engaged in the task. The mindfulness group demonstrated an average counting accuracy of 73%, aligning with the findings of [55] ([55]). The control group performed a 15 min free reading task. The reading material is a self-help manual focusing on mental health specifically designed for college students. Participants are permitted to freely explore any chapter of the manual within a designated timeframe. During the reading session, they are prohibited from engaging in conversations with others or participating in unrelated activities.

Habituation Phase: The large and small circles were each presented six times in a pseudo-random manner. During this phase, no electric shocks were administered following the presentation of the circles.

Acquisition Phase: The CS+ and CS− stimuli were presented 12 times each in a pseudo-randomized sequence, ensuring that no stimulus appeared consecutively more than twice. There was a 75% probability that an electric shock followed the CS+, while no electric shock was ever associated with the CS−.

During the habituation and acquisition phases, each trial adhered to a specified sequence: Initially, a red “+” is displayed at the center of the screen for a duration ranging from 14 to 16 s. Subsequently, the CS stimulus and detection interface are presented simultaneously, during which participants are instructed to evaluate the likelihood of the US using a numeric keypad. The CS stimulus remains visible for a maximum of 5 s, irrespective of whether the participants respond. If the stimulus image was linked to the US, an electric stimulus was triggered 200 ms prior to the image’s disappearance; thereafter, both the stimulus image and the US disappeared concurrently. The presentation process for the experimental stimuli is illustrated in Figure 1.

### 2.5. Data Analysis

A 2 × 2 repeated measures ANOVA was employed to analyze shock expectancy and SCRs during the habituation and acquisition phases. In this analysis, the groups of mindfulness and free reading were considered as between-subjects variables, whereas the stimulus types CS+ and CS− were identified as within-subjects variables. Bayes factor (BF) was used as a complement to the null hypothesis significance test (NHST, [54], [53]). Statistical analyses were conducted using *R* 4.3.2 and *JASP* 0.18.3 software.

## 3. Results

The descriptive statistical results are presented in Table 1. There were no significant differences between the BCT group and free reading group regarding state anxiety (*t* (61) = 0.02, *p* > 0.05, Cohen’s *d* = 0.005, *BF*_10_ = 0.26), trait anxiety (*t* (61) = −0.52, *p* > 0.05, Cohen’s *d* = −0.12, *BF*_10_ = 0.29), or mindfulness attention awareness (*t* (61) = 0.12, *p* > 0.05, Cohen’s *d* = 0.03, *BF*_10_ = 0.26) before the experiment.

### 3.1. Habituation

The results showed the main effect of the group on SCRs is not significant, *F* (1, 61) = 1.31, *p* > 0.05, *η*^2^*_p_* = 0.02, *BF*_incl_ = 0.65. Similarly, the main effect of stimulus is not significant, *F* (1, 61) = 0.93, *p* > 0.05, *η*^2^*_p_* = 0.02, *BF*_incl_ = 0.29, nor a significant interaction effect between group and stimulus, *F* (1, 61) = 0.25, *p* > 0.05, *η*^2^*_p_* = 0.004, *BF*_incl_ = 0.29. The above results indicate that before the acquisition of fear, there were no significant differences in fear responses to CS+ and CS− between the mindfulness group and the control group.

### 3.2. Acquisition

This analysis focused on participants’ shock ratings and SCRs during the acquisition phase. Furthermore, to explore differences in conditioned fear acquisition across groups, the difference scores between the CS+ and CS− conditions (CSD) across groups during the acquisition phase ([21]).

#### 3.2.1. US-Expectancy Ratings

The results indicated that the Bayes factor demonstrated a non-significant main effect of group, *F* (1, 61) = 5.56, *p* < 0.05, *η*^2^*_p_* = 0.08, *BF*_incl_ = 1.15. Additionally, the main effect of stimulus material was significant, *F* (1, 61) = 121.99, *p* < 0.001, *η*^2^*_p_* = 0.67, *BF*_incl_ = 2.60 × 10^19^. Furthermore, a significant interaction effect was observed between the group and stimulus material, *F* (1, 61) = 5.90, *p* < 0.05, *η*^2^*_p_* = 0.09, *BF*_incl_ = 5.66.

A simple effect analysis of the expected value of electric shocks indicated that both the mindfulness group (*t* (61) = 9.76, *p* < 0.001, Cohen’s *d* = 1.71, *BF*_10_ = 1.25 × 10^11^) and the control group (*t* (61) = 5.95, *p* < 0.001, Cohen’s *d* = 1.01, *BF*_10_ = 802.89) demonstrated significant fear acquisition. Nevertheless, a significant difference was observed between the mindfulness and control groups on the CS− (*t* (61) = 3.53, *p* < 0.001, Cohen’s *d* = 0.58, *BF*_10_ = 37.53), while no significant difference was found on the CS+ (*t* (61) = −0.24, *p* > 0.05, Cohen’s *d* = −0.04, *BF*_10_ = 0.26). Additionally, results from an independent samples t-test indicated a significant difference in CSD during the acquisition phase between the groups (*t* (61) = 2.43, *p* < 0.05, Cohen’s *d* = 0.62, *BF*_10_ = 2.94), with the mindfulness group demonstrating a significantly higher CSD compared to the control group (Figure 2a).

The above results indicated that both the mindfulness group and the control group acquired fear associated with shock expectancy. However, the fear acquisition was significantly more effective in the mindfulness group compared to the free reading group. These differences primarily manifested in the mindfulness group displaying a markedly lower shock expectancy for the safe signals (CS−) compared to the control group.

#### 3.2.2. Skin Conductance Responses

For the SCRs, the analysis revealed a significant main effect of group, *F* (1, 61) = 5.75, *p* < 0.05, *η*^2^*_p_* = 0.09, *BF*_incl_ = 3.08. The effect of stimulus materials was also significant, *F* (1, 61) = 102.73, *p* < 0.001, *η*^2^*_p_* = 0.63, *BF*_incl_ = 9.92 × 10^19^. However, the interaction effect between the group and stimulus materials did not reach significance, *F* (1, 61) = 0.001, *p* > 0.05, *η*^2^*_p_* = 0.001, *BF*_incl_ = 0.26.

Further analysis of the SCRs indicated that both the mindfulness group (*t* (61) = 7.35, *p* < 0.001, Cohen’s *d* = 1.29, *BF*_10_ = 2.77 × 10^7^) and the free reading group (*t* (61) = 6.70, *p* < 0.001, Cohen’s *d* = 1.29, *BF*_10_ = 1.24 × 10^4^) exhibited significant fear acquisition. Additionally, results revealed no significant difference in CSD during the acquisition phase between the two groups to be significant (*t* (61) = 0.01, *p* = 0.99, Cohen’s *d* = 0.002, *BF*_10_ = 0.26) (Figure 2b).

The above results revealed that participants in the mindfulness group demonstrated significantly lower SCRs to both the CS+ and CS− in comparison to those in the free reading group. Nonetheless, there were no significant differences in fear acquisition effects between the two groups, as both effectively developed fear responses.

## 4. Discussion

This study aimed to investigate the influence and mechanisms of BCT on fear acquisition using a discriminative fear conditioning paradigm. Those results revealed that the mindfulness group demonstrated significantly enhanced fear acquisition in comparison to the free reading group regarding shock expectancy. Specifically, the shock expectancy for the CS− in the mindfulness group was significantly lower than that of the free reading group, whereas no notable difference was found for the CS+. Concerning SCRs, the mindfulness group demonstrated a significantly reduced fear response to the CS relative to the free reading group. However, there was no substantial difference in SCRs between the two groups during fear acquisition.

### 4.1. BCT Increases Sensitivity to Conditioned Fear Acquisition

The capacity to identify potential threats in the environment and respond with suitable defensive behaviors is a survival mechanism that has evolved in humans ([20]). Recent studies examining the effect of mindfulness on fear acquisition have presented a range of perspectives, highlighting the complexity and controversy surrounding this topic. Research conducted by [23] ([23]) indicates that mindfulness practice can maintain individuals’ sensitivity to fear acquisition, suggesting that individuals in a mindful state possess a heightened awareness of potential threats, which facilitates the acquisition of fear-related information. Conversely, other studies have reached contrasting conclusions, indicating that mindfulness training does not markedly influence fear acquisition, which suggests that mindfulness practice may be insufficient to modify the learning processes when individuals encounter fear-inducing stimuli ([4]). Furthermore, research utilizing the classical conditioning paradigm has shown that mindfulness training can delay the onset of the first conditioned response and reduce the frequency of conditioned responses, thereby impairing the ability to learn fear ([19]).

Mindfulness training is a technique that enhances psychological resilience and coping abilities by fostering individuals’ awareness and acceptance of their present experiences. This study employs the BCT to investigate the impact of short-term mindfulness training on fear acquisition. The results indicate that following a single 15 min session of mindfulness breathing and counting training, participants demonstrated a significant increase in sensitivity to fear acquisition. Specifically, the findings support the research conducted by [23] ([23]), confirming the critical role of mindfulness training in heightening individuals’ sensitivity during fear acquisition, thereby promoting survival and adaptability. This result, indeed, contradicts the findings of [4] ([4]) and [19] ([19]). One potential explanation for this discrepancy relates to differences in experimental design across the studies, while another may involve the interference of extraneous factors during prolonged mindfulness training. For instance, during the extended period of mindfulness practice, ensuring the quality of the training becomes challenging, and participants are often influenced by various life events, which can further affect the experimental outcomes. Furthermore, the results illustrate that effective interventions are not limited to long-term mindfulness practices; individuals who undergo a single short-term mindfulness training session can also exhibit improved attentional focus. This enhanced attention enables them to filter out irrelevant information more efficiently and strengthens their capacity to perceive and identify potential environmental threats, thereby amplifying sensitivity to fear acquisition ([9]). This discovery underscores the significance of brief mindfulness training in enhancing human cognition and emotional regulation. Furthermore, it provides behavioral evidence suggesting that mindfulness training may improve the capacity to regulate attention to sensory inputs by strengthening the synergy between the pre-hippocampal region and the sensory cortex ([46]).

### 4.2. BCT Enhances Individuals’ Ability to Differentiate Safety Signals

Individuals with anxiety often encounter challenges in the learning process, particularly in associating CS− with a sense of safety; this association is frequently less effective ([37]). Consequently, they maintain elevated levels of fear responses or persistent anxiety, even in the presence of safety signals. In other words, anxious individuals struggle to suppress their emotional fear and anxiety, even in objectively safe environments. Supporting this phenomenon, [11] ([11]) illustrated that anxious individuals fail to differentiate safety from threat signals through typical learning processes, leading to feelings of stress and unease even in potentially safe situations. Thus, learning to recognize safety signals as a form of conditioned inhibition is vital for mitigating excessive fear responses in individuals with anxiety disorders ([42]). Through conditioned inhibition, individuals can gradually learn to associate specific signals with safety, effectively reducing the incidence of fear responses.

This study revealed that participants in the mindfulness group exhibited significantly lower shock expectancy for the conditioned stimulus that signifies safety (CS−) compared to the free reading group. However, no significant difference in shock expectancy for the conditioned stimulus that signifies danger (CS+) was observed between the two groups. These results indicate that mindfulness-based cognitive therapy (BCT) primarily enhances individuals’ ability to recognize safety signals, thereby facilitating the acquisition of conditioned fear. Specifically, BCT training can significantly reduce individuals’ fear of safety signals, which in turn diminishes excessive anxiety responses to non-threatening environments or events, ultimately lowering the risk of pathological anxiety. This finding provides valuable insights into the mechanisms by which mindfulness training regulates individual fear and anxiety responses and elucidates the persistent inconsistencies found in research on fear acquisition interventions. For instance, [52] ([52]) demonstrated that attachment-priming interventions effectively inhibited fear acquisition, whereas the present study indicated that short-term mindfulness training could facilitate it. The discrepancy in outcomes may be attributed to the mindfulness intervention’s promotion of the ability to discern safety signals (CS−), in contrast to the attachment priming approach, which suppresses responses to danger signals (CS+).

### 4.3. The Enhancement Effect of BCT on Fear Acquisition Shows a Dissociation Between Physiological and Subjective Indicators

[30] ([30]) proposed a two-system framework, asserting that the neural activity associated with fear in the brain operates through two distinct pathways. The fear excitation pathway allows fear-related information to bypass higher-level cortical monitoring, directly projecting from the brainstem to regions involved in fear excitation. This pathway primarily encompasses structures such as the amygdala, insula, caudate nucleus, and thalamus, which relay excitatory signals to the autonomic nervous system, leading to rapid fear activation predominantly reflected in physiological indicators ([41]; [51]). Conversely, the fear inhibition pathway transmits fear-related information to higher-order brain centers, facilitating a more nuanced evaluation and comparison. This pathway primarily includes regions such as the hippocampus, ventromedial prefrontal cortex (vmPFC), and anterior cingulate cortex (ACC), with fear assessment processes in these areas being primarily indicated by explicit measures such as subjective prediction values ([29]).

This study demonstrated that, while the overall fear response to the CS in the mindfulness group was significantly lower than that in the free reading group, no significant difference in fear acquisition was observed between the two groups. This finding indicated a dissociation between SCRs and subjective expectancy ratings. The emergence of this phenomenon may be linked to the neural activity pathways activated by mindfulness training. Previous research has shown that mindfulness meditation enhances activation in prefrontal brain regions associated with attention, which aids in reducing attention avoidance and improving the efficacy of attention allocation ([16]). Even brief mindfulness meditation training can engage the attentional network, including parietal and prefrontal structures, thereby augmenting activation in areas responsible for sustaining and monitoring attention ([12]). According to the two-system framework of fear, mindfulness training stimulates regions that coincide with the fear inhibition pathway, predominantly influencing explicit measures, specifically subjective expectancy ratings. However, some studies have challenged the dual-process model ([13]), and opinions diverge on the dissociation between subjective expectancy ratings and skin conductance levels ([40]). Consequently, caution is necessary when employing the aforementioned analysis to interpret the dissociative phenomenon observed in the experimental results.

### 4.4. Significance and Prospects

If individuals fail to recognize that a safety signal no longer indicates a threat, it can lead to a persistent fear response, which is a core pathological factor in anxiety-related disorders. The results of this study found that BCT enhances the discrimination of safety signals, improving individuals’ sensitivity in the acquisition of conditioned fear. This finding further elucidates the conflicting results associated with the effects of mindfulness training on fear acquisition and offers valuable insights for the prevention of anxiety spectrum disorders. In other words, short-term mindfulness training represented by BCT can reduce individuals’ fear responses to safety signals, allowing them to acquire fear responses that match the level of danger of the stimuli, thus improving their healthy adaptability to the environment. This study presents several limitations. First, because the sample comprised solely healthy college students, caution is warranted when generalizing and applying the results to clinical populations. Second, the study relied exclusively on the accuracy of BCT as an indicator for measuring mindfulness effects, and it lacked follow-up assessments of the effectiveness of the mindfulness training conducted during the intervention phase. Future research should enhance inter-group control of mindfulness interventions and further investigate whether BCT can improve the ability to discriminate generalized stimuli, thereby validating its inhibitory effects on fear generalization and elucidating its underlying mechanisms. Furthermore, due to the complexity of the relationship between mindfulness training and fear acquisition, additional studies are required to verify and explore the effects and mechanisms of mindfulness training on fear acquisition.

## 5. Conclusions

The conclusions of this study are as follows: (1) BCT training effectively enhances the sensitivity to the acquisition of conditioned fear. (2) The mechanism through which BCT facilitates this enhanced acquisition involves the inhibition of excessive fear responses to safety signals. (3) Furthermore, the mechanism underlying the promotion of conditioned fear acquisition through BCT training may be linked to the activation of frontal brain regions associated with attention.

## Figures and Tables

**Figure 1 behavsci-15-00263-f001:**
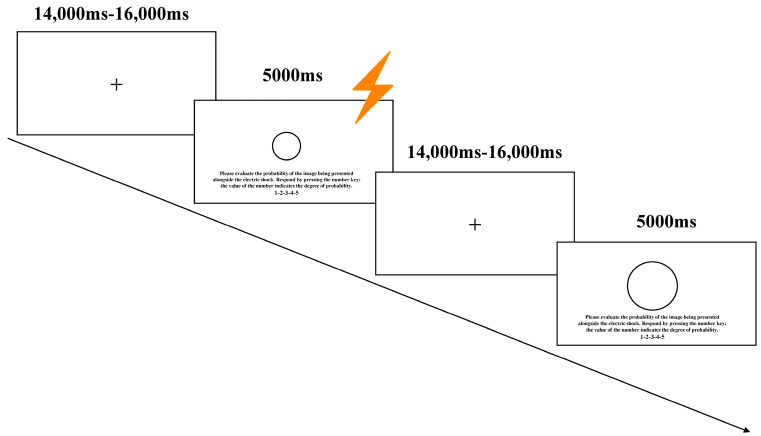
Schematic of fear conditioning and trial design. The conditioned stimulus materials were balanced across participants. The diameter of the large circle is 11.75 cm, and the diameter of the small circle is 5.00 cm.

**Figure 2 behavsci-15-00263-f002:**
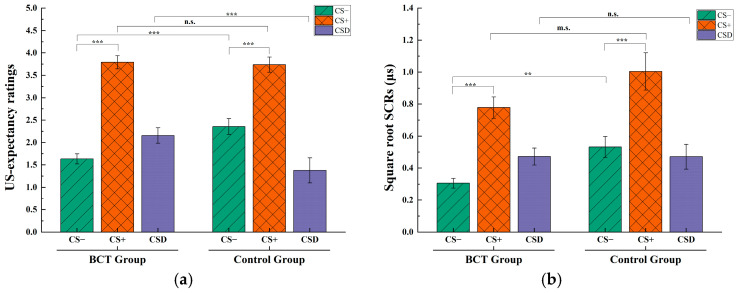
(**a**) US-expectancy ratings of different experiment groups; (**b**) SCRs of different experiment groups. The interaction effect between the group and stimulus materials on the SCRs is not significant. ** *p* < 0.01, *** *p* < 0.001. Bars represent the standard error of the mean. m.s. represents the marginal significance of the difference (*p* < 0.10), and n.s. represents no significant difference (*p* > 0.05).

**Table 1 behavsci-15-00263-t001:** The descriptive statistical results.

Group	SAI	TAI	MAAS	Measures	Habituation		Acquisition	
					CS−	CS+	CS−	CS+
BCT	1.96 (0.47)	2.14 (0.38)	3.02 (0.80)	Shock ratings	-	-	1.63 (0.65)	3.79 (0.84)
				SCRs	0.38 (0.26)	0.34 (0.29)	0.31 (0.18)	0.78 (0.39)
Control	1.96 (0.45)	2.20 (0.43)	3.00 (0.71)	Shock ratings	-	-	2.36 (0.96)	3.74 (0.93)
				SCRs	0.45 (0.34)	0.44 (0.32)	0.52 (0.36)	1.00 (0.64)

The data in the table are presented as mean (standard deviation). BCT refers to the mindfulness breathing group, while Control denotes the free reading group. SAI refers to the score of the State Anxiety Inventory, TAI refers to the score of the Trait Anxiety Inventory, and MAAS denotes the score of the Mindful Awareness Attention Scale. *N* = 63.

## Data Availability

The data of this study are available from the corresponding author upon reasonable request.

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
