# Peer review of "Breath-Counting Task Enhances the Sensitivity of Fear Acquisition"

_behavsci, 2025, doi:10.3390/bs15030263_

Round 1

Reviewer 1 Report

Comments and Suggestions for Authors

The article is an innovative article on short-time mindfulness interventions and fear learning, which may have important clinical implications in using mindfulness for people with several mental health symptoms. However, there are several things in the introduction and the discussion that should be clarified to strengthen the foundation of the study and make the flow of the manuscript better comprehensible for the reader.

Below, I am outlining some points to be clarified:

Introduction: 

·  Clarify the Gap in Research

  • The introduction acknowledges inconsistencies in previous findings on mindfulness and fear acquisition. Could the authors elaborate on the specific theoretical or methodological gaps their study aims to address?
  • Between line 61 and 107 the literature review should be written in a more consistent way, better identifying the gap and be less confusing 

·  Define the Relationship Between Mindfulness and Fear Learning

  • While the introduction discusses mindfulness broadly, it would be helpful to clarify why and how mindfulness might influence fear acquisition, beyond its general effects on attention and emotion regulation.
  • What specific cognitive or neural mechanisms support the hypothesis that mindfulness enhances signal discrimination in fear learning?
  • Authors mention anxiety-related disorders, but it might be worth brining also some literature related to trauma and PTSD. It is typical for people with trauma histories to perceive more sounds of danger, rather than safety. Does that mean that mindfulness may not be suitable for these individuals, since it may exacerbate their sensitivity to perceive threat? That would contradict the authors’ suggestions that mindfulness can be used with anxiety related disorders

·  Justify the Focus on Short-Term Mindfulness Training

  • The introduction highlights the challenges of long-term mindfulness interventions (e.g., time commitment, dropout rates), but it would be useful to discuss why short-term training is expected to be effective for fear acquisition.
  • Also, the authors cite MBSR and MBCT, which are the most evidence-based mindfulness interventions. What is the evidence behind short-term interventions? How do we know that using a short intervention makes empirical sense?
  • Is there empirical evidence supporting the efficacy of brief mindfulness interventions in modulating fear learning?

·  Explain the Choice of the Breath-Counting Task (BCT)

  • The introduction presents BCT as both a measure and a training tool for mindfulness, but further justification for why this specific task was chosen over other short mindfulness exercises such as the 3 min breathing space, squared breathing, mindfulness of touch, etc.
  • Does prior research suggest that BCT is particularly suited for influencing fear learning mechanisms and anxiety related disorders?

Methods, participants: 

·  Justification for Sample Size

  • The study initially estimated 34 participants based on a GPower calculation, but 70 participants were actually recruited.
  • Why was the sample size more than doubled? Was this an intentional adjustment, or was it to account for expected exclusions?
  • How does the final sample size (63 participants) affect the statistical power of the study?

·  Randomization Process

  • The participants were randomly assigned to groups, but was randomization stratified (e.g., by gender, age) to ensure balance between conditions?
  • Were there any significant differences between groups in terms of baseline characteristics?

·  Participant Demographics

  • The study specifies age, gender, vision, and mental health history, but were other potentially relevant factors (e.g., prior mindfulness experience, trait anxiety, trauma history, stress levels) assessed?
  • Why these inclusion criteria were selected?

·  Exclusion Criteria and Data Quality

  • Seven participants were excluded due to equipment malfunctions and careless responses in BCT—how was carelessness in BCT determined?
  • Did exclusions affect the balance between groups?
  • Were additional data quality checks performed (e.g., attention checks or response consistency measures)?

Discussion

Overall, the discussion lacks a clear hierarchical structure—key points sometimes feel buried in dense paragraphs. A clearer organization, with distinct sections for key findings, mixed results, theoretical implications, and practical clinical applications, would improve readability. In particular: 

Clarify the Main Contribution of the Study

  • The discussion covers multiple findings, but it could be clearer about the key takeaways.
  • What is the primary contribution of this study to the field of mindfulness and fear acquisition?
  • The authors could briefly summarize the most important finding in the first paragraph before breaking down the details.

Address the Mixed Findings More Critically

  • The study finds that mindfulness (BCT) enhances sensitivity to fear acquisition, which supports some prior research but contradicts others.
  • At the same time, it looks like authors are implying mindfulness both enhances and impairs fear learning across studies – the discussion should include a more clear explanation of why both are happening, what would that mean
  • Does the type of mindfulness intervention (short-term BCT vs. traditional MBSR/MBCT) explain the different effects?

Provide More Insight on the Practical Implications

  • The authors state that the findings could help prevent anxiety disorders, but how practically applicable is this short-term intervention?
  • If BCT improves safety signal recognition, could it be integrated into exposure therapy for anxiety?
  • Also, the authors may provide some more specific examples on how this intervention can be applied in mental health treatment and on different types of anxiety disorders – particularly, if individuals with histories of traumaare actually already hyper sensitive to fear stimuli, wouldn’t then mindfulness be even more “harmful” to them, as it will exacerbate the perceived fear response?

Good luck! 

Reviewer 2 Report

Comments and Suggestions for Authors

see attached file

Comments on the Quality of English Language

English language:

-          Ln 73 – “In the recent years” “in on the impact”

Overall the english language is of good quality. However there are some grammatical errors found in the Methods as listed below with suggestions for changes in [parentheses].

Ln 74 – “of mindfulness on the [ process of] fear acquisition and [fear] extinction

-       Method – change language to be consistent in past tense, e.g. “is [was] displayed” (line 156) for describing the procedural steps included in the experiment. Use present tense for describing tables, figures, etc included in the present article, e.g. Ln 204 – “was [is] illustrated in Figure 1.”

Round 2

Reviewer 1 Report

Comments and Suggestions for Authors

thank you for addressing my comments, I approve the publication of the article.